# Ionophore Ability of Carnosine and Its Trehalose Conjugate Assists Copper Signal in Triggering Brain-Derived Neurotrophic Factor and Vascular Endothelial Growth Factor Activation In Vitro

**DOI:** 10.3390/ijms222413504

**Published:** 2021-12-16

**Authors:** Irina Naletova, Valentina Greco, Sebastiano Sciuto, Francesco Attanasio, Enrico Rizzarelli

**Affiliations:** 1Institute of Crystallography, National Council of Research—CNR, Via Paolo Gaifami 18, 95126 Catania, Italy; irina.naletova@ic.cnr.it; 2National Inter-University Consortium Metals Chemistry in Biological Systems (CIRCMSB), Via Celso Ulpiani 27, 70126 Bari, Italy; 3Department of Chemical Sciences, University of Catania, Viale Andrea Doria 6, 95125 Catania, Italy; vgreco@unict.it (V.G.); ssciuto@unict.it (S.S.)

**Keywords:** carnosine, copper signaling, BDNF, VEGF, ionophore

## Abstract

l-carnosine (β-alanyl-l-histidine) (Car hereafter) is a natural dipeptide widely distributed in mammalian tissues and reaching high concentrations (0.7–2.0 mM) in the brain. The molecular features of the dipeptide underlie the antioxidant, anti-aggregating and metal chelating ability showed in a large number of physiological effects, while the biological mechanisms involved in the protective role found against several diseases cannot be explained on the basis of the above-mentioned properties alone, requiring further research efforts. It has been reported that l-carnosine increases the secretion and expression of various neurotrophic factors and affects copper homeostasis in nervous cells inducing Cu cellular uptake in keeping with the key metal-sensing system. Having in mind this l-carnosine ability, here we report the copper-binding and ionophore ability of l-carnosine to activate tyrosine kinase cascade pathways in PC12 cells and stimulate the expression of BDNF. Furthermore, the study was extended to verify the ability of the dipeptide to favor copper signaling inducing the expression of VEGF. Being aware that the potential protective action of l-carnosine is drastically hampered by its hydrolysis, we also report on the behavior of a conjugate of l-carnosine with trehalose that blocks the carnosinase degradative activity. Overall, our findings describe a copper tuning effect on the ability of l-carnosine and, particularly its conjugate, to activate tyrosine kinase cascade pathways.

## 1. Introduction

β-alanyl-l-histidine [1], a natural dipeptide named l-carnosine is synthesized by a cytosolic enzyme, carnosine synthetase [2]. In mammals, Car is mainly found in skeletal muscle, but it is also present in mM concentration in the olfactory bulb [3]. This endogenous dipeptide is a pH buffering agent [4,5] and protects cells from ROS, RNS and RCS insults by means of the histidine imidazole ring and the amino terminus of β-alanine residue. Namely, Car acts as a scavenger of superoxide radical anion, hydroxyl and peroxyl radicals, usually forming molecular adducts [6,7], and shows antioxidant activity against ^1^O_2_ [8]. In cell-free assay, Car shows a specific ability to reduce NO concentration in comparison with the constituent amino acids or their mixture. This direct scavenging feature against RNS results in the formation of carnosine/NO and carnosine/NO_2_ adducts [9]. Car also prevents the aggregation of different proteins, directly interfering with the non-covalent interactions between the polypeptide chains of the monomeric species that drive the oligomerization process [10,11,12,13]. This natural dipeptide displays multifunctional activities in vitro and in vivo [14,15,16,17,18,19,20,21,22,23,24,25,26,27,28,29,30]. Due to the above-cited pleiotropic abilities, Car and its analogues are called enigmatic peptides [31]. While the dipeptide protective function can be connected with its components in cell-free and biological assays, the potential therapeutic ability against every diverse disease cannot be attributed to a single biochemical pathway as related to antioxidant activities [32]. Thus, the suggested molecular mechanisms do not allow for overcoming the attribution of enigmatic peptides.

Car forms different complex species with copper(II) and zinc(II) ions [33], and its chelating ability provides different protective functions in vitro and in vivo [34,35,36,37,38]. Despite this relevant evidence, the Car–metal complex activity mechanisms are still poorly defined. In contrast with the suggestion of the dipeptide as a Cu^2+^ sequestering molecule [36], carnosine’s most noteworthy finding is its recent copper ionophore ability. Car cellular incubation induces a copper level increase in the cytosol in the culture medium; at the same time copper extracellular addition tends to lower further metal cellular uptake [39].

Ionophores and ionophorous agents were terms coined in 1967 [40] to generically classify antibiotics as alkali ions binding ligands able to rapidly spread through cellular membranes. Since the late sixties, the ionophore ability has been found in small molecules capable of binding d-block metal ions, (Cu, Zn, Fe, Co, Mn) and of facilitating their transport across cellular membranes [41]. Ionophores [42,43], therefore, can affect the cellular metallome [44], modifying the overall metal ion distribution among the different cellular compartments and directing them where they are needed [45]. In addition, this class of metal ligands can change the concentrations of intracellular specific metal species inducing a partitioning between tightly bound metal pools buried within proteins (the so-called static metallome) [46] and those not tightly bound to proteins (dynamic metallome, labile or exchangeable metal ions) [47].

Metal-chaperones [48], transporters (SLCs) [49], transcription factors (TFs) [50], storage molecules, such as metallothioneines (MTs) [51], and low molecular weight molecules like glutathione [52] belong to the dynamic metallome and represent the cellular network that ensures the metallostasis (metal homeostasis) and the redox ho-meostasis of the cells [53]. This complex architecture is better known for zinc and cop-per.

The members A1 (Ctr1) and A2 (Ctr2) of the SLC31 family of solute carriers l [54] belong to a protein network regulating the intracellular Cu^+^ level within a certain con-centration range. Ctr1 is a protein with high specificity and affinity for Cu^+^, and the most important protein that transports copper into the cell [55].

Inside the cell, chaperones drive Cu^+^ to specific partners, namely i. the superoxide dismutase copper chaperone (CCS) that transports the metal ion to Cu, Zn superoxide dismutase (SOD1); ii. Cytochrome c oxidase 17 (COX17) that mediates copper incorpo-ration into cytochrome c oxidase; and iii. antioxidant 1 (ATOX-1) that delivers copper to two P-type ATPases, A and silica gelB. [56,57].

The ionophore ability of Car affects the expression of two key players of the metallostasis network, that control intracellular copper level, represented by the Slc31a1 transmembrane Cu-importer Ctr1 [58] and the related Cu-responsive tran-scription factor Sp1 [50]. But equally worth of note are the remarkable findings con-cerning Car ability to permeate the blood brain barrier and induce the secretion of some neurotrophins (NTs) [59] in glial cells [60]. Furthermore, the dipeptide stimulates brain functions and curbs cognitive decline through brain-gut interaction, by activa-tion of the transcription factor, cAMP response element-binding protein (CREB) [61]; CREB in turn, as main regulator of Brain Derived Neurotrophic Factor (BDNF), induces its synthesis and release also from intestinal epithelial cells [62,63].

BDNF is required for brain normal development [64,65] and is involved in synaptic plasticity, long term potentiation, learning and memory, neurogenesis and axonal regeneration [66,67,68,69]. The synthesis of BDNF is characterized by a multistep process that requires the formation of different precursor isoforms; like the other NTs, it is firstly obtained as pre-pro-BDNF [70]; the precursor proBDNF undergoes cleavage intra- or extra-cellularly [71] to produce a mature mBDNF protein that is released together with proBDNF. While proBDNF preferentially binds p75 NTR receptor, which induces apoptosis [72], the secreted mBDNF binds specifically to tyrosine kinase receptors (TrkB) and promotes cell survival [73]. Furthermore, the binding of mBDNF to TrkB receptor gives rise to its dimerization and autophosphorylation [74]; phosphorylated TrkB is able to activate different kinase cascade pathways: phosphatidylinositol 3-kinase (PI3K), mitogen-activated protein kinase (MAPK) and phospholipase C-c (PLC-c) [75,76]. The activated signaling cascades involve: (i) the PI3K/Akt-related pathway [75,77]; (ii) the MAPK-MEK1/2/-Erk1/2 related signaling [78]; and, (iii) PLC-c-dependent pathway which activates Ca2+-calmodulin-dependent protein kinase and protein kinase C [76]. All these pathways converge on CREB, a major mediator of neuronal neurotrophin responses [79] Small peptides encompassing amino acid residues of various NT domains show signaling pathways like those induced by cognate ligands [80], while linear peptide fragments of the neurotrophin N-terminus, in the presence of copper and zinc, reproduce the signaling effect of NTs, including activation of tyrosine kinase cascade. However, they do not just mimic the NT mode of action [81,82]. The reported ionophore ability of these peptides suggests an inhibitory effect on tyrosine phosphataseB [83]; however, the underlying mechanisms remain unknown.

Recent studies report that different classes of small molecules are able to release copper and zinc intra-cellularly, inducing metal-dependent PI3K, MAPK and matrix metalloprotease (MMP) activation and decreasing oligomeric amyloid species in AD models in vitro. Furthermore, the cellular uptake of zinc and copper in different brain areas leads to activation of different biochemical pathways, including tyrosine kinase cascades [84,85,86]; these findings reinforce the function, recently attributed to copper, of intracellular signaling regulator [87,88,89,90].

Having in mind the dual Car ability to act as copper ionophore and to induce NT expression and release by means of CREB, here we demonstrate the role played by Car-assisted copper to activate tyrosine kinase cascade pathways in PC12 cells, stimulating the expression of the trophic factor BDNF.

Furthermore, because of the suggested connection [91] between neurotrophic and angiogenic factors based on BDNF and vascular endothelial growth factor (VEGF) [92,93] in relation to neurogenesis [94], the sharing of the two trophic factors of the same family of tyrosine kinase receptors [95], and the role of copper in angiogenesis, we also studied the ability of Car to favor copper signaling inducing the expression of VEGF [96,97].

However, it is well known that the potential therapeutic action of Car is drastically hampered by its hydrolysis due to the serum [98,99] and tissue [100] carnosinase enzymes [99,101]. Conversely, Car derivatives and its conjugates with different polysaccharides are able to block or delay the Car degradation [102,103,104]. Hence, we also describe the properties of a conjugate (Tre-Car) obtained by the covalent binding of Car to trehalose (Tre); this disaccharide protects Car from hydrolysis induced by carnosinase [105], potentiates the protective functions of the dipeptide and, analogously to Car, forms complex species with copper(II) ion [106].

Finally, clioquinol (CQ, 5-chloro-7-iodo-8- hydroxyquinoline), a small hydrophobic molecule that shows copper and zinc moderate affinity and transports these metal ions inside the cells, [107,108,109] was also shown to be a comparable ionophore able to affect cell signaling pathways [110,111].

## 2. Results

### 2.1. The Trehalose Moiety in Tre-Car Does Not Alter Car Effect on Cell Viability, While It Potentiates the Intracellular Uptake and Ionophore Ability of the Natural Dipeptide

PC12 cells have been widely used to investigate multiple neurobiological processes, including neuronal differentiation, intracellular signaling pathways, and cell survival [112].

To study the cytotoxic/proliferative effect of Tre, Car, Tre-Car, Tre + Car and CQ, PC12 cells were treated for 24 h, and then the viable cells were quantified by MTT assay (Figure 1). Since copper is usually present at physiological conditions in the culture medium (µM level) [113], and Car incubation in murine B104 cells increases Cu intracellular uptake from the culture medium [39], all cellular experiments were carried out without adding copper ion. Furthermore, the results (Figure 1) were compared with those obtained when cells were treated with 50 µM membrane-impermeable copper chelator BCS. The dose-response results indicate that the trehalose conjugate with Car does not alter the cell viability up to a concentration of 10 mM, and BCS does not modify this behavior.

Considering that studied Car compounds act as ionophores that transport copper into cells increasing its intracellular level, clioquinol, that can coordinate copper with high affinity [114], was used as a positive control in some assays. Figure 1 shows no significant changes in cell viability after the addition of 10 µM CQ.

Afterwards, the effect of 24 h treatment with extracellular Car, Tre-Car and the Tre + Car mixture on intracellular accumulation of Car was estimated by indirect ELISA assay (Figure 2). The cell lysate level of the dipeptide, revealed by using a carnosine-specific antibody, increased up to 183 ± 2% and 201 ± 7% in Car or Tre-Car treated cells, respectively, as compared to the untreated control cells. Interestingly, the presence of 50 µM BCS does not affect intracellular accumulation of Car. The trehalose conjugation with the dipeptide favors the Car ability to pass through the cellular membrane, further increasing its level inside the cell.

### 2.2. Car and Tre-Car Perturbs Ctr1 and Sp1 Levels Differently, Indicating Dissimilar Ionophore Ability

Recent results suggest that Car can favor the cellular uptake of copper present at micromolar level in culture medium [39]. To ascertain the ionophore ability of Tre-Car in comparison with Car, the high-affinity copper transporter 1, Ctr1 and the transcriptional factor Sp1 expression levels in culture of PC12 cells were investigated. It is well known that copper level increase induces a Ctr1 protein internalization [115], regulating, therefore, metal homeostasis; then, the protein recycles back to the membrane after copper removal, suggesting a post-translational mechanism of Ctr1 expression in response to copper stress.

Figure 3A shows the decrease of the Ctr1 level after the treatment with Tre-Car (52 ± 12%), Car (74 ± 15%) and the Tre + Car mixture (81 ± 13%); the presence of Tre alone does not affect, as expected, the copper transporter expression due to its inability to act as metal ligand. The presence of BCS in the medium affects the expression of Ctr1, whose level increases up to 202 ± 21% due to the sequestering of extracellular copper, present in the medium, by the chelator, and the consequent reaction of the metal uptake membrane transporter to counteract the cellular copper deprivation. The Ctr1 increase level does not change in the case of combined presence of Tre and BCS; conversely, Car (164 ± 11%), Tre-Car (145 ± 19%) or Tre + Car (145 ± 16%) contribute to decrease of the BCS effect in an almost similar way. Car or Tre-Car act as different copper(II) ion competitor ligands for BCS; they maintain a certain ability to partly transport the metal ion inside the cell that decreases the level of homeostatic reaction to activate the Ctr1 expression. This trend suggests a thermodynamic control on the different expression level of Ctr1 due to the difference between the high affinity (log K_11_ = 12.42) of copper complexes with BCS [116] in comparison with that of Car (log K_11_ = 8.49) [117] and Tre-Car (log K_11_ = 7.27) [106]; in addition, the different ability of the dipeptide and its derivative to permeate the cellular membrane can contribute to overall results. Analogously, the differences in copper affinity gradient can be invoked to explain the trend found in the absence of BCS. Recently, the affinity constant of the ATCUN copper(II) ion binding site, present in the Ctr1 ectodomain, has been reported (log ^c^K_7.4_ = 13.0) [118]; this high affinity constant drives the more favorable uptake of the metal ion bound to Tre-Car whose stability constant is lower than that of the analogous complex with Car, making the conjugate ligand a less competitive system than Car in transferring the metal ion to Ctr1 and, therefore, inducing the consequent different expression of the membrane transporter. CQ does not affect the expression of Ctr1 (96 ± 22%); the more stable metal complex of CQ (log K_11_ = 10.8) [119] in comparison with that of Car accounts for its lesser effect on Ctr1 level.

The up- and down-regulation of mammalian copper homeostasis by Ctr1 is regulated by the transcription factor Sp1, that tunes the copper transporter 1 under metal ion perturbed conditions [120]. Figure 3B shows that after 24 h of treatment, the Sp1 expression is significantly down-regulated only by Tre-Car (83 ± 9%) that is able to transport into the cell the copper ions present at µM levels in the culture medium. In line with such findings, our results show that the expression of Sp1 after the treatment with BCS increased up to 175 ± 17%, whereas the concomitant incubation in the presence of Car, Tre-Car, or Tre + Car mixture induces significant decrease of BCS effect, up to 132 ± 15%, 120 ± 13%, and 137 ± 11%, respectively. As in the case with Ctr1 expression, CQ under the used concentration, does not affect the expression of Sp1 (106 ± 11%).

All these findings indicate that the investigated ligands are able to transfer copper into the cell, affecting Ctr1 and Sp1 that act as related sensors of copper homeostasis, evidencing that Tre-Car is the best ionophore consistent with its lowest copper affinity constant.

### 2.3. Copper Signaling Assists Car and Tre-Car Induction of CREB Phosphorylation and Its Modulators Erk and Akt

CREB is a stimulus-induced transcription factor [121] that is involved in the transcription of more than 4000 genes [61] in response to different arrays of stimuli [122], including those provided by peptide hormones, trophic factors and the protein kinase family. Specific amino acid residue phosphorylation (serine 133) is needed for CREB-mediated transcription [122].

The dose dependent effect of Tre, Car, Tre-Car Tre + Car mixture and 10 µM CQ on CREB phosphorylation at Ser-133 was investigated. Protein extracts were obtained from PC12 cells exposed to 1, 5 or 10 mM of studied compounds for 30 min (Figure 4). Car increases pCREB level vs. total CREB and reveals a significant dose-dependent effect on CREB phosphorylation, up to 162 ± 6% and 181 ± 8% at 5 and 10 mM concentrations, respectively. Moreover, Tre-Car (1, 5 and 10 mM) induces phosphorylation of CREB (117 ± 4%, 209 ± 11% and 256 ± 11%, respectively) significantly more than Car alone, while the incubation with trehalose slightly increases the level of pCREB up to 129 ± 4% only at the 10 mM. Considering the ability of Tre-Car to strongly increase phosphorylation of CREB as compared to Car or Tre alone, we incubated cells with Car and Tre mixture. The cell treatment by the mixture stimulates CREB phosphorylation to the level of 155 ± 11% and 181 ± 10% at 5 and 10 mM, respectively; these values are comparable with those determined by Car alone. Like Car and Tre-Car, CQ also favours CREB phosphorylation up to 130 ± 8%. A recent study reports that Car (10 mM) also activates CREB phosphorylation in Caco-2 cells [63] but it occurs less efficiently than in used here PC12.

Therefore, taking into consideration the dose-dependent (Figure 4) and the cell viability (Figure 1) results, a 5 mM concentration was employed in the following assays to overlook the potential contribution of Tre that is irrelevant at this concentration.

The addition of BCS pre-treated medium alone does not affect pCREB, while it induces a significant decrease of CREB phosphorylation as compared to the non-pretreated medium, for both Car (from 170 ± 14% to 132 ± 8%), Tre-Car (from 235 ± 22% to 200 ± 10%) and Tre + Car mixture (160 ± 8% to 119 ± 4%) (Figure 5). The chelating ability of BCS counteracts the ionophore effect of Car and Tre-Car, suggesting that the phosphorylation process is dependent on the signal from intracellular copper transferred by Car or its conjugate from the medium into the cells as evidenced by the Ctr1 and Sp1 changes above reported.

As previously mentioned, Car (10 mM) is able: (i) to activate CREB and CREB related pathways, including BDNF expression and release, by activating Ca^2+^-related pathways in Caco-2 cell line, and (ii) to increase CREB-dependent genes in the intestine, contributing to the peptide-induced activation of brain–gut interaction [63]. Our findings suggest a nonexclusive different pathway due to the significant contribution of the copper binding ability of the dipeptide and its conjugate associated with their ionophore ability, and the copper ion signaling effects on protein kinase cascade pathways.

Binding of the cognate ligands with their RTKs triggers the MEK/Erk and PI3K/Akt protein kinase signaling pathways that are key regulators of CREB activation [123]. Different studies alter this scenario due to the ability of copper ions to stimulate PI3K signaling [84] in a way independent not only by the production of reactive oxygen species [124] but also by the involvement of cognate ligands of receptor tyrosine kinases [125]. In addition, a specific player of copper ion homeostasis, Ctr1, is reported to be needed to signal activation of the MAPK and PI3K pathway by the ligands of three major RTKs, including FGF, PDGF and EGF [123]. Interestingly, the cellular ablation of Ctr1 impair the phosphorylation of both Erk1/2 and Akt suggesting that the Cu-dependent step is upstream of the point at which the Ras/Raf/MEK/Erk and Ras/PI3K/Akt pathways differ [123]. Further involvement of copper in this protein kinase pathways is supported by the finding concerning, both, the direct interaction of metal ion with MEK1 in vitro and the reduction of phosphorylation processes by copper extracellular chelating agents such as TTM (tetrathiomolybdate) and BCS [123].

The effect, therefore, of Car, Tre, Tre-Car and Tre + Car mixture on Erk1/2 and Akt phosphorylation was examined to highlight the involvement of copper ion present in the culture medium (Figure 6 and Figure 7). Car alone (183 ± 9%), in the mixture with Tre (202 ± 5%) or Tre-Car (366 ± 46%) induces significantly Erk1/2 phosphorylation (Figure 6). The phosphorylation of Akt (Figure 7) on Ser-473, in response to the stimulus by Car (195 ± 18%), Tre-Car (368 ± 39%), and Tre + Car mixture (194 ± 24%), follows a pattern similar to the one observed for CREB and Erk1/2. Moreover, coherently with CREB phosphorylation, both Erk1/2 and Akt phosphorylation results in less activation in the presence of BCS. Noteworthy, Tre-Car displays a stronger effect than Car alone on protein tyrosine kinase cascade activation; this fact can be related to the different ability of the investigated chelating molecules to act as ionophores. The treatment of the cells with 10 µM ionophore CQ, both in the case of Erk1/2 and Act, showed an increase in phosphorylation up to 143 ± 20% and 135 ± 22%, respectively.

Altogether, the findings indicate a sort of “menage a trois” of the intracellular copper ion, the dipeptide (or its conjugate) and the protein kinase Erk1/2 (or Akt) that connects a d-block metal signal to classical kinase signal pathways.

### 2.4. BDNF Expression and Release as Well as VEGF Expression Are Induced by Copper Signaling in Presence of Car and Tre-Car

Recent results indicate that Car induces expression and secretion of NGF and BDNF in U-87 MG cells, but not in SH-SY5Y cells, suggesting that the natural dipeptide activates neuronal cells through increased production of neurotrophins in glial cells [60] which act as ‘copper-sponge’ [126]. Keeping in mind the cited findings and the phosphorylation effect on the tyrosine kinase proteins Akt, Erk1/2 and the transcription factor CREB, the study was extended to BDNF expression and release, to further assess the signaling role of copper to drive the ionophore effect of Car, Tre-Car and the mixture of Tre + Car. Furthermore, being aware that BDNF acts both as a signaling partner of VEGF in angiogenic tube formation, and stimulator of angiogenesis through a VEGF signaling pathway [127], while Cu^2+^ induces the proliferation of endothelial cells [128] and promotes wound healing by upregulation of VEGF [129], the Car and its conjugate effect on this trophic factor was also studied.

Figure 8 shows the increase of BDNF release (A) and expression (B) in the medium and cell lysates, respectively, after 24 h of treatment. Using ELISA BDNF kit, we analyzed BDNF release in the culture medium. The incubation of cells with the studied compounds increase the BDNF production: Car (136 ± 3%), Tre-Car (145 ± 9%) and Tre + Car mixture (136 ± 7%). On the other hand, by western blotting an increase of mature BDNF can be observed in the cell lysate upon incubation with Car (150 ± 2%), Tre-Car (198 ± 6%) and Tre + Car mixture (153 ± 12%) (Figure 8B). It is important to note that incubation with CQ, on one hand, leads to increase in Erk1/2 and Act phosphorylation (Figure 7), but, on the other hand, has no effect on protein BDNF expression (Figure 8).

BCS appears to slightly decrease the effect on BDNF. This unusual behavior in comparison with other results presented here deserves some comments. Extracellular copper(II) ion can be portioned between the ionophore ligands and BCS as above discussed, with the phenanthroline derivative able to sequester Cu^2+^ according the higher affinity constants. In this assay, the BCS ability to bind Cu^2+^ significantly decreases suggesting the presence of another competitor ligand. Recently, the BDNF N-terminus peptide was reported to bind copper(II) ion [130], thus the secreted BDNF can compete with BCS for metal ion and in some way preserve the copper signal. In addition, the involvement of BDNF receptor can be invoked due to a new hypothesis that suggests a Cu^2+^ direct “activation of RTK signaling probably via the enhanced dimerization between monomer RTKs”, in addition to intracellular downstream stimulation of Erk1/2 and Akt [131].

Car shows an antiangiogenic activity associated with its antitumor effect against different cancers [132]; conversely, a preventive treatment with Car of hypoxia-induced neurotoxicity in rats (250 mg/kg) increases hypoxia inducible factor-1α (HIF-1α), VEGF, and its receptor VEGFR1 stimulating angiogenesis [133]. Furthermore, Car supplementation shows enhanced post ischemic hind limb revascularization by the increase of HIF-1α angiogenic signaling and its pretreatment of murine myoblast (C2C12) cells enhances HIF-1α protein expression, VEGF mRNA levels and VEGF release under hypoxic conditions [25]. The hypothesis suggests that a Car complex with iron(II) ion could stabilize HIF-1α against degradation action of prolyl hydroxylase domain protein, favoring its translocation to nucleus with transcription of proangiogenic genes such as VEGF [134].

VEGF expression is increased up to 259 ± 4% after the treatment with Tre-Car, with the conjugate being more active than Car alone (208 ± 2%) or Tre + Car mixture (218 ± 6%) (Figure 9); the trend is the same as found for BDNF release and the results can be related to the different ionophore ability of the different molecules. The involvement of copper signal is clearly indicated by the significant decrease of the trophic factor by adding BCS. At the same time, the effect of CQ on VEGF expression is different from that for BDNF. Treatment of cells for 24 h resulted in an increase in VEGF level in cells of up to 147 ± 7%.

Altogether, the findings highlight the role of copper in connecting BDNF and VEGF release and expression; this interplay is further supported by the ability of Cu^2+^ to interact with the VEGFR1 domain2, inducing dimer formation [135] and recalling the effect of VEGF interaction with its receptor. Furthermore, it is noteworthy that both the specific receptors of BDNF and VEGF belong to the same family of RTKs and their ectodomains share a significant copper(II) ion affinity with subsequent activation of extracellular metal signaling.

The involvement of HIF-1α can also be explained by the ability of copper to stabilize this transcription factor [136] activating VEGF expression.

### 2.5. Ionophore Ligands and Cu^2+^ in Culture Medium Do Not Significantly Affect GSH Redox State

ROS are common subproducts of oxidative energy metabolism and are a significant physiological modulators of several intracellular signaling pathways, including the MAPK [137] and PI3K [138] pathways. Cu^2+^ exposure to cells involves its reduction to Cu^+^ and transport into the cell by means of Ctr1; intracellular Cu^+^ may interact with molecular oxygen (present at concentration 10^−5^ M), to form different ROS and cause Erk1/2 and Akt activation. In order to ascertain if this pathway is responsible for the effect of our compounds, the level of glutathione in its reduced form, which is the most abundant intracellular reductant, was determined. Figure 10 shows that added copper (50 μM) decreases the level of reduced GSH due to ROS production that is responsible for GSH thiol oxidation. Conversely, the treatment with Tre, Car, Tre-Car or Tre + Car mixture with or without BCS does not alter the GSH/GSSG ratio. The results suggest that the transfer of culture medium copper into the cell activates the above reported protein kinase cascades without the involvement of oxidative stress stimuli.

## 3. Discussion

Although many studies report on the protective effect of carnosine in diabetes, cancer, neurodegeneration and other diseases [139], the mechanisms of action are still to be elucidated, while antioxidant, oxygen free-radical scavenger, anti-glycation and antiaggregant, pH buffer and metal chelating activities are related to the structural features of the different moieties encompassing the dipeptide [3,27,140].

Specifically, l-carnosine chelating ability accounts for the metal removing activity to counteract copper overload in cells [141] or to inhibit copper-driven amyloid-β protein aggregation and oxidative stress [142]. Furthermore, the dipeptide acts as ion chelator displaying neuroprotective activity against Cu^2+^-driven neurotoxicity [35], both reverting the inhibitory action of extracellular copper on P2X4 and P2X7 receptors and removing the metal already bound to these receptors [36].

Although, the reported studies highlight the metal removing ability of Car, the description of either physiological interplay or crosstalk between this natural dipeptide and Cu^2+^ is still inadequate.

Our findings show that carnosine and its conjugate not only permeate the cellular membrane increasing the intracellular peptide level (Figure 2) but also enhance the copper content inside the cell (Figure 3A); BCS counteracts this last effect blocking the metal extracellular pool uptake. Consistently, the changes in Ctr1 and Sp1 expression levels testify the ability of the metallostasis network to appreciate the low variation of copper level as the one caused by the partial influx of micromolar metal amounts present in the culture medium (Figure 3) [143]. This ionophore effect of the studied molecules that show a moderate copper affinity [106,117] recalls what was found for MPAC ligand, a term coined to indicate “organic small molecule that, without the high affinities of chelators, can ligate adventitial metal ions related to proteinopathies and redistribute them to a safer compartment” [144].

This new perspective of Car function resembles what experienced by CQ, a prototypic MAPC molecule that, initially utilized as sequestering agent of copper and zinc from Aβ aggregates [145], successively showed an ionophore ability transporting zinc and copper into cells [108] and restoring intracellular metallostasis. Though both Car [60] and CQ [146] share the ability to cross the brain blood barrier, their efficacy in transport of copper inside the cell is different (Figure 3). The ionophore effect of Car is bigger than that of CQ, and it is evidenced by the different decrease in Ctr1 level.

The comparison between Car and Tre-Car highlights the trehalose role in favoring the copper uptake by the carnosine conjugate that amplifies the slight difference in the ability to transfer the cellular membrane between the two molecules (Figure 2). Tre, furthermore, appears as a non-innocent moiety for Car, not only protecting it from carnosinase effect but also affecting its ionophore effect.

Different studies indicate Ctr1 that provides most of intracellular copper uptake as membrane transporter of Cu^+^, as a main player in activating metal signaling by the in-cell labile copper pool. Different reports highlight that Ctr1 is involved in the activation of signaling by the kinase pathways with the cognate ligands of different receptor tyrosine kinases which represent the most abundant family of enzyme-linked receptors characterized by extracellular ligand binding domain with endogenous protein tyrosine kinase activity in cell receptors [147]. The mechanism by which Ctr1 exerts this effect is through its control of the availability of intracellular Cu^+^ that induces the phosphorylation of Erk1/2 and/or Akt through different receptor tyrosine kinases.

Different scenarios describe the ability of copper signaling to activate tyrosine kinase pathways [131]. Cu^2+^ (50–100 μM) interaction with the epidermal growth factor receptor (EGFR) and other RTKs upregulates the receptor phosphorylation in metal-dose-dependent manner with downstream activation of intracellular kinase pathways and increase of Akt and Erk1/2 in a time-dependent mode.

Our findings show that Ctr1 is involved in conjugate ability of Car to assist copper signaling responsible for the kinase pathway activation. The low micromolar copper, present in the culture medium, partly is transported into cells and stimulates not only the Erk and Akt activity but also the CREB phosphorylation. BCS decreases the copper signaling reducing intracellular metal uptake.

Partly consistent with this behavior of Car is the one showed by CQ, whose copper complexes specifically activate EGFR by a ligand independent phosphorylation of the receptor by means of src-kinase and downstream activation of Erk but not of Akt [148].

Recent results attribute to Car the ability to activate CREB pathway in Caco-2 cells [63]; our findings suggest that the role of the peptide is to assist the stimulation activity of copper ions in the phosphorylation process of the transcription factor [61]. Downstream CREB activation, two different mechanisms of Car-induced activation of neuronal cells are reported, both related to the involvement of NTs, including BDNF. According to the first mechanism, Car activates intestinal epithelial cells and causes them to secrete trophic factors that reach the brain and activate brain function [62], while the second mechanism of carnosine activation of brain function is based on the ability of the peptide (1–10 mM) to permeate the blood brain barrier and activate glial cells causing the secretion of NGF, BDNF, NTF4 and GDNF in human primary glioblastoma cell line(U-87 MG) [60].

Our results show the ionophore ability of Car and its conjugate to activate the copper signaling to induce BDNF expression/secretion utilizing the low concentration of metal ion in culture medium, suggesting that the main player in BDNF expression is copper, as supported by the BCS induced changes (Figure 8B).

This ligand independent activation of BDNF by copper is consistent with previous results showing that CuSO_4_ (1 μM) added to culture medium is able to induce a 3-fold increase in the BDNF mRNA level as compared to untreated control in PC12 cells [81]. The presence of a peptide fragment encompassing the NGF N-terminus causes a further increase of BDNF expression by an ionophore effect.

Usually, BDNF interacts with its receptor, TrkB, inducing phosphorylation which leads to different signals including differentiation, plasticity, and survival of neuronal cells [149]. Different studies highlight the role of copper signaling in BDNF activity: CuCl_2_ (10 µM) both induces tyrosine phosphorylated TrkB in a metalloproteinase-assisted mode and activates BDNF signaling facilitating the conversion from pro-BDNF to mBDNF [130].

VEGF tightly controls angiogenesis by the binding and the activation with tyrosine kinase receptors VEGFR-1 and VEGFR-2; recent studies also evidence that VEGF is a potent trophic factor [93]. At the same time NGF and BDNF are shown to act as remarkable angiogenic growth factors [150] that can tune the angiogenic response by giving rise to VEGF expression [151]. Recent studies report a strict interconnection between BDNF and VEGF in neurotrophic and antidepressant-like activities [91] showing that: (i) BDNF-TrkB binding induces VEGF release; (ii) neurotrophic activity of BDNF requires VEGF-VEGFR2 signaling; and (iii) VEGF stimulates BDNF release [151]. This partnership between the two trophic agents is evidenced during fracture healing where BDNF regulates the expression and secretion of VEGF from osteoblasts by the activation of TrkB/Erk1/2 pathway [152].

A player highlighting the interplay between BDNF and VEGF is the Hypoxia-inducible factor 1a. HIF-1α is not only the upregulated transcription factor consequent to the activation of BDNF/TrkB signaling that increases VEGF mRNA and protein expression [153] but also the upstream regulator of VEGF [154]. Furthermore, different studies report that copper is required for activation and regulation of transcriptional complex formation of HIF-1 involving also CCS, a copper chaperone of -Cu/Zn-superoxide dismutase [136,155].

In the context of previous discussion, copper primarily can connect two trophic factors; it promotes angiogenesis stimulating VEGF production and wound healing [129], while the underlying mechanism involves the transcriptional activity of HIF-1α that modulates VEGF expression [156,157].

Inhibitors of protein tyrosine kinase activities nullify CuSO_4_-driven VEGF protein as well as mRNA expression [127], suggesting an involvement of the metal ion in the specific tyrosine kinase receptor activation. This finding suggests a behavior parallel to that reported above concerning Cu^2+^ ability to interact with the ectodomain of TrkB; it is noteworthy that BDNF and VEGF receptors both belong to the RTK family. Namely, both extracellular domains show similar structural features with the presence of histidine residues which contribute to the cognate ligand binding and can also act as chelating anchors for d-block metal ions such as copper(II) and zinc(II). Consistently, a recent report shows that human VEGFR1 domain 2 crystallizes in the presence of Cu^2+^ as a dimer that is formed via metal ion interactions and interlocked hydrophobic surfaces, recalling the dimerization effect induced by VEGF interaction with its receptor [135]. A competition test puts in evidence that Cu^2+^ is able to displace VEGF from its VEGFR1 extracellular domain at micromolar level. All these findings highlight the signaling effect of copper in its two redox states, Cu^2+^ at extracellular site by interaction with receptor ectodomains and Cu+ inside the cell by means the PI3K/Akt, ERK1/2, and CREB activation with a (reasonably conceivable) HIF-1α contribution. Actually, a previous report shows that CQ, as a Car analogous Cu^2+^ ionophore, induces functional HIF-1α activation and its downstream VEGF [158] expression. Our results show that Car and, particularly, Tre-Car increase VEGF expression, but this effect is decreased in the presence of BCS highlighting the role of copper.

Overall, our findings describe a copper tuning effect on the ability of carnosine and its conjugate with trehalose to activate tyrosine kinase cascade pathways. Further studies are needed to assign the contribution of copper signaling to the biological pleiotropic effects of this natural peptide and its derivatives, as well as to its potential therapeutic activities.

## 4. Materials and Methods

### 4.1. Chemicals

Anhydrous α,α-Trehalose, silica gel 60 F254 plates, l-carnosine, iodine, triphenyphosphine, acetyl chloride, anhydrous dimethylformamide and anhydrous methanol were purchased from Merck KGaA, Darmstadt, Germany.

### 4.2. Synthesis of Tre-Car

The synthesis of conjugate Tre-Car has already been reported in a patent [105] wherein 6-bromo-6-deoxy-threhalose has been used as a synthetic intermediate; we deemed it appropriate to revisit the synthesis at that time proposed in order to better suit our subsequent synthetic needs, in terms of operational viability and yields.

To this end, in order to optimize the coupling step between the carnosine unit and the halogen derivative of trehalose, we replaced the intermediate 6-bromo-6-deoxy-threhalose with the corresponding iodinated peracetylated compound (6-iodo-6-deoxy-peracetylthrehalose). Considering that alkyl iodides give the highest yields in nucleophilic substitution, this would allow the Tre-Car conjugate to be obtained in more satisfactory yields.

Therefore, the selected synthetic route required the preparation of 6-Iodo-6-deoxy-peracetylthrealose [159,160], its subsequent coupling with the methyl ester of carnosine, the removal of all protecting groups and the final purification of the obtained conjugate (Figure 1).

### 4.3. Cell Line

Rat pheochromocytoma (PC12) cells were obtained from the American Type Culture Collection (Manassas, VA, USA) and cultured at 12 passages number in PRMI-1640 medium supplemented with 10% horse serum (HS), 5% fetal bovine serum (FBS), 2 mM l-glutamine. Cell culture was grown in a humidified atmosphere of air/CO_2_ (95:5) at 37 °C in an incubator Hera Cell 150 (Heraeus, Germany).

### 4.4. Cytotoxicity Assays

The cytotoxicity of Tre, Car, Tre-Car and Tre + Car mixture in the presence or in the absence of 50 µM BCS ((2,9-Dimethyl-4,7-diphenyl-1,10-phenanthroline disulphonic acid) was tested on PC12 cell culture at 60–70% of confluence. Cells were seeded at a density 2 × 10^4^cells per well in 200 µL complete medium on l-poly-lysinated 96 tissue culture plates (Eppendorf, Eppendorf North America, Inc., Enfield, CT, USA). Compounds, prepared as stock solutions in ultrapure water Milli-Q, were added for 24 h in complete RPMI-1640 medium supplemented with 1% HS and 0.5% FBS. Cell viability was determined at 37 °C by the 3-(4,5-dimethylthiazol-2-yl)-2,5-diphenyltetrazolium bromide (MTT) method. Reaction was stopped with DMSO and the absorbance was measured at 569 nm by plate reader (Varioskan^®^ Flash Spectral Scanning Multimode Readers, Thermo Scientific, Waltham, MA, USA). Results were expressed as % of viable cells over the concentration of each compound and presented as the means ± SD; the experiments were performed 3–4 times in triplicate.

### 4.5. Enzyme Linked Immuno Sorbent Assay (ELISA)

Cells were treated 24 h with threhalose, carnosine, trehalose-carnosine or trehalose and carnosine with or without 50 µM BCS in RPMI-1640 medium supplemented with 1% HS and 0.5% FBS. Afterwards, medium and cells were collected and examined.

Conditioned medium was diluted with the provided sample buffer (1:1) and immediately prior to the assay. Level of BDNF in PC12 cells supernatants was measured with Rat BDNF ELISA Kit (RAB1138, Sigma-Aldrich, St. Louis, MO, USA).

Cell lysates were harvested with RIPA buffer (50mM TRIS-HCl, pH 8.0, 150 mM NaCl, 0.5 mM EDTA, 1% Triton X-100) containing Halt Protease and Phosphatase Inhibitor Single-Use Cocktail (ThermoFisher, Waltham, MA, USA). The concentration of carnosine in cell lysates was determined by indirect ELISA assay. PVC microtiter plate was coated during 2 h at 37 °C with cell lysates diluted with carbonate/bicarbonate buffer (pH 9.6). Then plate was washed with PBS, blocked with blocking buffer (5% non-fat dry milk/PBS) for 1 h at room temperature and incubated with primary anti-carnosine antibody (Cat#PAB0084, 1:500 dilution) overnight at 4 °C. After, plates were washed with PBS, incubated for 2 h with goat anti-rabbit IgG secondary antibody horseradish peroxidase-conjugated (Cat#AP307P, 1:3000 dilution) and washed again with PBS. Result was detected by 3,3′,5,5′-tetramethylbenzidine (TMB) solution after the incubation for 15 min. The reaction was stopped by stop solution (2 M H_2_SO_4_) and the optical density was measured at 450 nm by plate reader (Varioskan^®^ Flash Spectral Scanning Multimode Readers, Thermo Scientific).

### 4.6. Western Blotting Analysis

For cell lysates analysis by western blotting, cells were seeded on 12 multiwell l-poly-lysinated tissue culture plates at a density 3 × 10^5^ cells per well in complete medium for 24 h until cellular adhesion was attained. Then cells were treated for every experimental point in triplicate for 15 or 30 min (analysis of phosphorylated proteins) or 24h (analysis of protein expression) with Tre, Car, Tre + Car mixture or Tre-Car in completed RPMI-1640 medium or in the medium pre-treated for 24 h with 50 µM BCS (experiments were repeated at least 3 times). Afterwards, cells treated in triplicate were collected in one eppendorf and then harvested with RIPA buffer containing Halt Protease and Phosphatase Inhibitor Single-Use Cocktail (Thermo Fisher). Lysates were separated by SDS-PAGE with 4–12% precast gel and transferred to nitrocellulose.

The transfers were blocked with blocking buffer at room temperature for 1 h, incubated with primary antibodies overnight at 4 °C. Anti-phospho-Akt (S473) (Cat#9271, 1:1000 dilution), anti-Akt (Cat#4685, 1:1000 dilution), anti-phospho-Erk1/2 (T202/Y204) (Cat#9106, 1:1000 dilution), anti-Erk1/2 (Cat # 9107, 1:1000 dilution), anti-phospho-CREB (S133) (Cat#9191, 1:1000 dilution), anti-CREB (Cat#9197, 1:1000 dilution) and anti-β-Actin (Cat#4970, 1:2000 dilution) antibody were from Cell Signaling Technology (Danvers, MA, USA). Anti-GRB2 antibody (Cat#sc-17813, 1:500 dilution) was purchased from Santa Cruz Biotechnology (Santa Cruz, CA, USA). Anti-CTR1 (Cat#ab129067, 1:3000 dilution), anti-SP1 (Cat#ab13370, 1:2000 dilution), anti-BDNF (Cat#ab108319, 1:2000 dilution) and anti-GAPDH (Cat#ab8245, 1:2000 dilution) were purchased from Abcam (Waltham, MA, USA). Anti-VEGF antibody (Cat#PAB12284, 1:1500 dilution) was purchased from Abnova Corporation (MA, USA). After, membranes were incubated 1h with goat anti-rabbit or anti-mouse antibodies, labeled with IRDye680 (1:20,000 dilution, LI-COR Biosciences, Lincoln, Nebraska USA), and hybridization signals were detected with the Odyssey Infrared Imaging System (LI-COR Biosciences). Western blot data were quantified by densitometric analysis of the hybridization signals from membranes obtained by more than three different independent experiments. The nitrocellulose membrane, if necessary, was then stripped with buffer Restore (Pierce, Rockford, IL, USA) and, subsequently, re-probed with the specific antibodies for the unphosphorylated total proteins. GRB2, actin and GAPDH were used as loading control. The level of phosphorylation was calculated as ratio between data from anti-phospho antibodies over those from the related not phosphorylated counterparts.

### 4.7. Glutathione Redox State Determination

Cells were incubated with Tre, Car, Tre-Car or Tre + Car mixture for 16 h, then trypsinized and counted to obtain a cell suspension of 8 × 105 cells/mL. In order to evaluate the ratio of GSH/GSSG we used the detection assay kit from Abcam (Cat#ab138881, Abcam, Waltham, MA, USA). Lysates were obtained by adding PBS with 0.5% of NP-40. Cells were homogenized by pipetting and centrifuged (10 min at 14,000 r.p.m.). Supernatants were treated by deproteinizing Sample Preparation Kit–TCA (Cat#ab204708, Abcam, Waltham, MA, USA) and were then collected and processed according to the kit protocol (excitation/emission wavelengths = 490/520 nm).

### 4.8. Statistical Analysis

Each data value is expressed as the mean ± SD (*n* = 3 or 4). Data were analyzed by Student’s *t*-test for multiple variable comparisons. A *p*-value of less than 0.05 or 0.01 or 0.001 is considered significant.

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
