# Peer review of "Ionophore Ability of Carnosine and Its Trehalose Conjugate Assists Copper Signal in Triggering Brain-Derived Neurotrophic Factor and Vascular Endothelial Growth Factor Activation In Vitro"

_ijms, 2021, doi:10.3390/ijms222413504_

Round 1

Reviewer 1 Report

This paper reports a study on the widely distributed in mammals natural dipeptide L-carnosine whose properties regard its antioxidant, anti-aggregating and metal chelating ability revealed in a large number of physiological effects.

The aim of this work is to contribute to the better knowledge of the biological molecular mechanisms involved in the protective role of L-carnosine found against several diseases.

Since, it was known that L-carnosine increases the secretion and expression of various neurotrophic factors and affects copper homeostasis in nervous cells, the authors report the copper binding and ionophore ability of L-carnosine and its trehalose conjugate to activate tyrosine kinase cascade pathways in PC12 cells and stimulate the expression of BDNF and VEGF.

The findings point out that L-carnosine and its conjugate (1) not only permeate the cellular membrane increasing the intracellular peptide level but also enhance the copper content inside the cell with consequence in the changes in Ctr1 and Sp1 expression levels; (2) are involved to assist copper signaling responsible for the kinase pathway activation. The copper present in the culture medium at low micromolar concentration, is partly transported into cells and stimulates not only the Erk and Akt activity but also the CREB phosphorylation; (3) have a copper tuning effect on the ability to activate tyrosine kinase cascade pathways.

Experiments are well done, and the conclusions are convincing. I consider this paper as an important contribution to the deciphering of the biological molecular mechanisms involved in the protective role of L-carnosine.

Author Response

We thank the reviewer for the positive evaluation of our manuscript. 

Reviewer 2 Report

In their manuscript, Ionophore ability of carnosine and its trehalose conjugate assists copper signal to trigger Brain Derived Neurotrophic Factor and Vascular Endothelial Growth Factor activation in vitro. The authors present an interesting investigation, which requires some revisions in order to become publishable.
Specific points
1. What does the "*" in Figure S1 represent, which seems to be out of place.
2. It is suggested to unify the positions of size labels in all protein maps, such as Figure 2, Figure 6 and Figure 7.

Author Response

We appreciate the reviewer’s suggestions and apologize for the inconvenience in the following figures.

Point 1: What does the "*" in Figure S1 represent, which seems to be out of place.

Response 1: We have replaced the figure S1 in supplementary information with the one where the asterisk is positioned correctly.

Point 2: It is suggested to unify the positions of size labels in all protein maps, such as Figure 2, Figure 6 and Figure 7.

Response 2: Considering the reviewer’s suggestion we have modified the figures legends (lines: 218, 262, 281, 323, 330, 363, 392) for better understanding. We added the following phrase: “Dashes to the right of the membrane bands show the positions of prestained molecular mass marker.”